# Feeding Frequency Affects the Growth Performance and Intestinal Health of Juvenile Red-Tail Catfish (*Hemibagrus wyckioides*) with the Same Amount of Daily Feed

**DOI:** 10.3390/ani15111621

**Published:** 2025-05-30

**Authors:** Baohong Xu, Zheyu Wen, Chen Zheng, Shengguo Tang, Tiaoyi Xiao, Yaqun Qiu, Qiaolin Liu

**Affiliations:** 1Fisheries College, Hunan Agricultural University, Changsha 410128, China; xbh_1012@hunau.edu.cn (B.X.); wzy13787721392@sina.com (Z.W.); tiaoyixiao@hunau.edu.cn (T.X.); 2Hunan Engineering Technology Research Center of Featured Aquatic Resources Utilization, Hunan Agricultural University, Changsha 410128, China; 3Hunan Provincial Key Laboratory of Water Pollution Control Technology, Hunan Academy of Environmental Protection Sciences, Changsha 410004, China; zhengchen200010@sina.com; 4Institute of Yunnan Circular Agricultural Industry, Pu’er 665000, China; tangshengguo2000@sina.com

**Keywords:** *Hemibagrus wyckioides*, growth performance, intestinal microbiota, metabolome, transcriptome, serum biochemical parameters, digestive enzyme activity, hepatic antioxidant parameter

## Abstract

Feeding frequency is a crucial factor in determining feed intake and plays an important role in controlling fish appetite and nutrient utilization. Therefore, it is a critical factor in aquaculture efficiency and productivity. In this study, through a 56-day feeding trial, we assessed the effects of feeding frequency on the growth, intestinal health, and metabolism of larval red-tailed catfish (*Hemibagrus wyckioides*). Our results showed that a feeding frequency of three times per day improves the growth performance, lipase activity, and relative abundance of beneficial *Clostridium* in gut microbiota. These results imply that a feeding frequency of three times per day is the optimal feeding frequency of larval red-tailed catfish. This study provides practical guidance and experimental data for *H. wyckioides* culture.

## 1. Introduction

In 2022, for the first time in history, aquaculture surpassed capture fisheries as the main producer of aquatic animals [1]. The global aquaculture production reached 130.9 million tonnes, with 94.4 million tonnes being aquatic animals, accounting for 51% of the total aquatic animal production [1]. As a result, aquaculture has become a crucial source of food supply [2,3]. The pursuit of optimal efficiency and productivity in aquaculture has always been a significant goal in aquaculture research [4].

Feeding frequency is a crucial factor in determining feed intake and plays an important role in controlling fish appetite and nutrient utilization [5,6]. Therefore, it is a critical factor in aquaculture efficiency and productivity. The optimal feeding schedule should consider the behavioral patterns and developmental stages of species [7]. The feeding habits and digestive tracts of aquatic animals affect their digestive efficiency and nutrient absorption when different feeding frequencies are used. Inappropriate feeding frequencies can impair growth, metabolism, activity of digestive enzymes, and digestive function in fish, and even lead to oxidative stress and reduced immunity, resulting in significant economic losses [8,9,10,11,12]. However, the underlying mechanisms by which feeding frequency influences fish metabolism and immune response are still unknown.

The red-tail catfish (*Hemibagrus wyckioides*) is a highly valued commercial fish species in the Lancang River Basin due to its rapid growth rate, high protein content, and strong disease resistance, making it a popular choice for consumption [13]. Research on *H. wyckioides* has primarily focused on nutritional factors, such as optimal dietary lipid levels and alternative sources of protein, such as fish meal [14,15], as well as breeding factors, including genetic diversity and QTLs for growth- and sex-related traits [12,16]. Currently, intensive aquaculture production of *H. wyckioides* is widespread in the Mekong River Basin, with the fish being farmed in ponds and cages [13]. The ideal feeding frequency is typically determined by the feed conversion rate, but there is limited information available on how feeding frequency specifically affects *H. wyckioides*.

Therefore, we conducted a study using hematological parameters, intestinal transcriptome, metabolome, and microbiota analyses to evaluate the impact of feeding frequency on *H. wyckioides*. Our results provide a scientific foundation for the efficient cultivation of *H. wyckioides*.

## 2. Materials and Methods

### 2.1. Test Design and Culture Conditions

The culture experiment was conducted at Yule Agriculture Co., Ltd. (Puer, China) using three-dimensional circulating water culture tanks. The juvenile *H. wyckioides* used in the experiment were provided by Xishuangbanna Fish Fry Breeding Co., Ltd. (Yunnan, China). Prior to the experiment, all fish were acclimated to the feeding conditions for a period of two weeks in a recirculating water system.

After acclimation, 450 fish (8.47 ± 0.36 g) were randomly distributed into three groups. Each group contained three tanks with a diameter of 2 m and a height of 1 m. Each tank contained 50 fish, and there were three replicates in each group (Appendix A). The fish were fed at different frequencies: twice per day (F2, at 8:00 and 20:00), three times per day (F3, at 8:00, 14:00, and 20:00), and four times per day (F4, at 8:00, 12:00, 16:00, and 20:00) for a total of 56 days during the culture experiment. The daily feeding amount for each fish was 3% of their body weight, divided equally among each meal according to the feeding frequency. The feeding amount was adjusted every ten days.

Commercial feed (Tongwei, Chengdu, China) for *H. wyckioides* (water ≤ 12%, crude fat ≥ 4%, crude ash ≤ 15%, and crude protein ≥ 36%) was used. The water conditions were as follows: pH, 7.4 ± 0.2; temperature, 21–24.6 °C; NH3-N concentration, <0.2 mg/L; salt concentration, <0.01 mg/L; and dissolved oxygen concentration, >5 mg/L.

### 2.2. Sample Collection

After the feeding trial, all fish from each test group were subjected to a 24 h fast. We then anesthetized them with 100 mg/L MS-222 (Sigma, St. Louis, MO, USA). Five fish from each tank were randomly selected and individually weighed. A total of 45 fish were analyzed (three groups with three tanks in each group). Because blood collection takes time, to minimize the influence of different storage times before serum biochemical parameter determination on the results and ensure sufficient sedimentation of plasma, blood samples were collected and stored for 12 h at 4 °C. They were then centrifuged at 3000 rpm for 10 min at 4 °C to obtain serum, which was subsequently stored at −80 °C. The liver and intestine were collected from five fish per tank, pooled from the same tissue source, and stored at −80 °C. Additionally, intestinal tissues were fixed with 4% paraformaldehyde for 24 h for histological analysis. The intestinal contents were also collected and stored at −80 °C for analysis of the intestinal microbiota composition.

### 2.3. Growth Performance and Composition Analysis

After the feeding trial, the fish were weighed in groups to assess their growth parameters, including the feed conversion ratio (FCR), specific growth rate (SGR), weight gain ratio (WGR), survival rate (SR), hepatosomatic index (HSI), viscerosomatic index (VSI), and condition factor (CF) [17].

The moisture, crude protein, crude lipid, and ash contents of the fish were measured using standard methods [18].

### 2.4. Serum Biochemical Parameter Determination

We evaluated the serum biochemical parameters using commercial kits of Nanjing Jiancheng Bioengineering Institute (Nanjing, China). These parameters included total cholesterol (TC), triglyceride (TG), high-density lipoprotein cholesterol (HDL-C), low-density lipoprotein cholesterol (LDL-C), glucose (GLU), total protein (TP), and albumin (ALB) levels.

### 2.5. Intestinal Digestive Enzyme Activities and Hepatic Antioxidant Parameters

The activities of intestinal trypsin, amylase, and lipase, as well as the hepatic antioxidant indices, including total antioxidant capacity (T-AOC), malondialdehyde (MDA), superoxide dismutase (SOD), glutathione peroxidase (GSH-PX), alanine transferase (ALT), alkaline phosphatase (ALP), and aspartate transferase (AST), were determined using commercial detection kits from Jiancheng Bioengineering Institute (Nanjing, China).

### 2.6. Morphology of Foregut Tissue

Small 1 cm foregut segments were excised from the specimens. Tissue samples (5 μm thick) were stained with hematoxylin and eosin (H&E), and then observed with microscope observation (Pannoramic DESK, P-MIDI, P250; 3D HISTECH, Budapest, Hungary). Image-Pro Plus^®^ 6.0 software was used to measure the intestinal villus height (Media Cybernetics, Silver Spring, MD, USA). Fifteen fish were analyzed in each group (five fish in each tank), and one section was taken for each fish, and three villi were measured in each section.

### 2.7. Reverse Transcription—Quantitative Real-Time PCR (RT-qPCR)

Total RNA of the liver tissue was extracted using TRIzol reagent (Invitrogen, Waltham, MA, USA). An established protocol [19] was followed, with β-actin as the internal reference gene. The expression level of the target genes was assessed using the 2^−ΔΔCT^ method [20]. The primer sequences are listed in Appendix A.

### 2.8. Profiling Hindgut Microbiome

Hindgut microbial DNA was extracted using an OMEGA soil DNA kit (Omega Bio-Tek, Norcross, GA, USA). The V3-V4 hypervariable region was amplified using the prokaryotic primer pair 338F (5′-ACTCCTACGGGAGGCAGCA-3′) and 806R (5′-GGACTACHVGGGTWTCTAAT-3′), as previously described [21]. High-throughput sequencing was performed on the intestinal microbiota 16S rRNA using an Illumina platform (Illumina, USA) at Personal Biotechnology Co., Ltd. (Shanghai, China) in paired-end mode. The TruSeq Nano DNA LT Library Prep Kit (Illumina, USA) was used to construct libraries. Amplicon sequence variants (ASVs) were generated by QIIME2 with the Greengenes database. Low-abundance ASVs were removed. The Bray–Curtis distance was used to measure β-diversity and was visualized through principal coordinates analysis (PCoA) and non-metric multidimensional scaling (NMDS). Bacteria with |log_2_FoldChange| > 1 and *p* < 0.05 were considered to have a significant difference in abundance. Four adjacency matrices of ASVs were determined using the Spearman correlation coefficient. Microbial networks were structured using RMT theory with a uniform similarity threshold of 0.94.

### 2.9. Intestinal Transcriptome Analysis

Total RNA was extracted from intestinal tissue using TRIzol reagent (Invitrogen, Waltham, MA, USA). The concentration and quality of the RNA were assessed using a NanoDrop spectrophotometer (Thermo Scientific, Waltham, MA, USA), and the integrity of the RNA was confirmed using a Bioanalyzer 2100 system (Agilent, Santa Clara, CA, USA). High-quality RNA samples were then used to construct sequencing libraries. A threshold of |log_2_FoldChange| > 1 and *p* < 0.05 were applied to identify differentially expressed genes. The KEGG pathways of these genes were enriched using Cluster Profiler v4.6.0.

### 2.10. Intestinal Metabolism Analysis

Nontargeted LC-MS was utilized to identify the intestinal metabolites of *H. wyckioides* fed at different frequencies, in both positive- and negative-ion modes, to investigate the effects of feeding frequency on the metabolic characteristics of the intestinal microbiota. Intestinal metabolites were extracted using methanol containing 5 ppm of 2-chlorophenylalanine. A portion of the extracted test sample was combined with quality control (QC) samples, whereas the remainder was analyzed using liquid chromatography-mass spectrometry (LC-MS) with an ACQUITY UPLCHSS T3 column (100 Å, 1.8 µm, 2.1 × 100 mm). MS data were acquired using an Orbitrap Exploris 120 MS (Thermo scientific, Waltham, MA, USA). All formal samples and QC samples were loaded into the machine according to the LC-MS methods described by Zheng et al. [22]. Prior to the formal injection, 2–4 QC samples were injected to balance the system. During the injection process, a QC sample was injected every 5–10 samples for subsequent data evaluation and quality control.

The raw data from the instrument were imported into the commercial software Compound Discoverer™ 3.3 version 3.3.2.31 (Thermo, Waltham, MA, USA) for peak extraction, alignment, and correction. Missing values for undetected peaks were filled in using the FillGaps algorithm, and the total peak areas were normalized. All metabolites were classified according to KEGG and MetPA. The mzCloud online database (https://www.mzcloud.org/, accessed on 28 December 2024), HMDB (https://hmdb.ca/, accessed on 28 December 2024), and MoNA (https://mona.fiehnlab.ucdavis.edu/, accessed on 28 December 2024) databases were utilized, with the NIST2020_SMS spectral library. The MS1 mass tolerance was set to 15 ppm, while the MS2 match factor threshold was set to 50.

### 2.11. Statistical Analysis

Statistical analyses were performed using SPSS Statistics 22.0 for Windows (IBM, Chicago, IL, USA) and SAS v.9.4 M3 (SAS, Cary, NC, USA). Statistical significance was determined by one-way ANOVA followed by Tukey’s post hoc test. Results are shown as the mean ± standard error, and significance was defined as *p* < 0.05.

## 3. Results

### 3.1. Growth Performance, Feed Utilization, and Body Composition

There were no significant differences in the SR (One-way ANOVA, *p* = 0.999), HSI (one-way ANOVA, *p* = 0.709), VSI (one-way ANOVA, *p* = 0.878), or CF (one-way ANOVA, *p* = 0.531) among the three feeding frequency groups of *H. wyckioides* (Table 1). However, when compared to the F4 group, the FCR in the F3 group was significantly lower (one-way ANOVA, *p* = 0.048), whereas the SGR (one-way ANOVA, *p* = 0.044) and WGR (one-way ANOVA, *p* = 0.043) were significantly higher (Table 1). No significant difference was observed when compared to the F2 group (one-way ANOVA, *p* > 0.05; Table 1).

There were no significant differences in whole-body composition among the different feeding frequency groups, including moisture (one-way ANOVA, *p* = 0.942), crude protein (one-way ANOVA, *p* = 0.901), crude lipid (one-way ANOVA, *p* = 0.548), and crude ash (one-way ANOVA, *p* = 0.713) contents (Table 1).

### 3.2. Serum Biochemistry

The serum TG concentration did not show a significant difference among the different feeding frequency groups (one-way ANOVA, *p* = 0.371). However, the serum TC concentration increased with feeding frequency, with the F4 group exhibiting a significantly higher concentration compared to the other groups (one-way ANOVA, *p* = 0.023). Additionally, the serum HDL-C concentration of the F3 group was significantly higher than that of the F2 group (one-way ANOVA, *p* = 0.049). The levels of TP (one-way ANOVA, *p* = 0.301), GLU (one-way ANOVA, *p* = 0.253), and ALB (one-way ANOVA, *p* = 0.673) in *H. wyckioides* were slightly higher in the F3 group compared to the other two groups (Table 2).

### 3.3. Liver Antioxidant Indices

The activities of liver T-AOC (one-way ANOVA, *p* = 0.026), SOD (one-way ANOVA, *p* = 0.025), GSH-PX (one-way ANOVA, *p* = 0.044), and MDA (one-way ANOVA, *p* = 0.001) were significantly higher in the F4 group compared to the other groups, whereas the liver ALT (one-way ANOVA, *p* = 0.001) and AST (one-way ANOVA, *p* = 0.053) activities in the F3 group were higher than those in the other groups (Table 3).

### 3.4. Digestive Enzyme Activity and Morphological Characteristics in Foregut

The activity of intestinal trypsin significantly increased with increasing feeding frequency (one-way ANOVA, *p* = 0.004; Table 4). However, there was no significant difference in the activity of intestinal amylase among the feeding frequency groups (one-way ANOVA, *p* = 0.937; Table 4). The activity of intestinal lipase in the F3 group was significantly higher than in the other groups (one-way ANOVA, *p* = 0.002; Table 4). Histological analysis revealed that the intestinal villi of *H. wyckioides* maintained their complete structure and clear margins across all feeding frequency groups, with consistent and orderly villus surface cell nuclei (Figure 1). No significant differences in the intestinal muscle thickness (MT) or villus length (VL) were detected among the groups (*p* > 0.05; Table 5).

### 3.5. Liver Antioxidant Genes

The expression levels of the *gpx* (ANOVA, *p* = 0.528; Figure 2A), *sod* (ANOVA, *p* = 0.396; Figure 2B), *cat* (ANOVA, *p* = 0.381; Figure 2C), *nrf2* (ANOVA, *p* = 0.488; Figure 2D), and *keap1* (ANOVA, *p* = 0.934; Figure 2E) genes did not differ significantly in the *H. wyckioides* liver among the different feeding frequency groups (Figure 2).

### 3.6. Hindgut Microbiota Analysis

A total of 488,574 high-quality sequences were obtained from the hindgut microbiota, with an average read count of 54,286 per sample. There were 332 bacterial ASVs detected among all the samples, and the number of ASVs from the F2, F3, and F4 groups was 136 (27.47%), 178 (35.96%), and 181 (36.57%), respectively (Figure 3A). Among these, 63 ASVs were present in all groups, and the number of ASVs specific to the F2, F3, and F4 groups was 51, 85, and 96, respectively (Figure 3A). Trimming and aligning sequences yielded a phylogenetic tree that covered 17 bacterial phyla, with Fusobacteriota, Proteobacteria, and Firmicutes being the most abundant (Figure 3A). The relative abundances of Fusobacteriota and Proteobacteria did not differ significantly among the different feeding frequency groups (Figure 3B). However, the relative abundance of Bacteroidota in the F2 group was significantly lower than that in the F4 group (*p* < 0.05, Figure 3B). The dominant genera were *Cetobacterium* (classified into Fusobacteriota), *Citrobacter*, *Plesiomonas*, *Aeromonas* (classified into Proteobacteria), *Clostridium*, *Lactococcus* (classified into Firmicutes), and *Paludibacter* (classified into Bacteroidota). Notably, the relative abundance of *Aeromonas* in the F3 group was significantly higher than that in the F4 group (*p* < 0.05, Figure 3C). The richness did not differ significantly among all the groups, although the F3 and F4 groups generally had a higher richness than the F2 group (*p* > 0.05, Figure 3D). The PCoA diagram based on Bray-Curtis distance accurately characterized community variations in the intestinal microbiota, as evidenced by the significant differences among all clusters (Figure 3E).

### 3.7. Differential Analysis and Mining Biomarkers Related to Specific Phenotypes

Compared to the F2 group, the intestinal microbiota of *H. wyckioides* in the F3 group showed 3 upregulated and 14 downregulated ASVs, whereas the F4 group had 25 upregulated and 19 downregulated ASVs (Figure 4A). In the F3 group, the upregulated ASVs were from Bacteroidetes and Proteobacteria, whereas the downregulated ASVs were mainly from Firmicutes and Proteobacteria (Figure 4B). In the F4 group, the upregulated ASVs were mainly from Proteobacteria and Actinobacteriota, whereas the downregulated ASVs were primarily from Proteobacteria (Figure 4B). A total of 48 distinct ASVs were identified from the three pairwise comparisons (Figure 4C).

Three modular eigenvectors (MEs) were identified and assigned taxonomic classifications based on the differential ASV abundance matrix. The phenotypic data matrix was divided into four eigengenes (Appendix A). Consequently, the feature modules “antioxidant_enzyme_gene1” and “antioxidant_enzyme_gene2” (which are related to the *H. wyckioides* phenotype matrix and consist of *cat*, *sod*, and *gpx*) showed a strong correlation with ME2 and ME3 of the differential ASV abundance matrix (Appendix A). Further analysis of ME2 and ME3 revealed three major microbial contributors: *Plesiomonas* (ASV60 and ASV66, assigned to Proteobacteria) and *Streptococcaceae* (ASV422, assigned to Firmicutes; Appendix A). Interestingly, the relative abundance of ASV60 was significantly lower in the F4 group compared to the F2 group, and the relative abundance of ASV66 was significantly lower in the F4 group compared to the F3 group (*p* < 0.05; Appendix A).

### 3.8. Molecular Ecological Network Analysis

Based on the variations in the assembly processes of fish intestinal microbiota due to different feeding frequencies, a classification model was developed and quantified. The results of neutral community modeling studies indicated that the microbiota composition was mainly influenced by deterministic processes (Appendix A).

All networks showed high modularity, with each one divided into six submodules. The intestinal microbiota of *H. wyckioides* in the F4 group exhibited the most complex network, with the highest numbers of nodes and edges, as well as the highest average degree. A total of 108 module pairs were calculated, and there were 18 single modules and 17 module pairs with significant differences across groups, accounting for 15.74% and clustering into three module clusters (Appendix A). The F4 group had a lack of modules classified into cluster four, and only a few modules were retained in clusters one and two. All networks showed a preference for coexistence rather than coexclusion, with positive correlations accounting for 83.35–92.86% of the potential interactions evaluated (as shown in the table of topological roles). The number of negative edges varied, with the lowest negative correlation (5.24%) observed in the network of the F4 group, whereas the F2 group had more negative interactions.

### 3.9. Intestinal Transcriptome

A total of 4518 differentially expressed genes (DEGs) were identified among the different feeding frequency groups. Specifically, there were 1810, 3853, and 597 DEGs between the F3 and F2 groups, the F3 and F4 groups, and the F2 and F4 groups, respectively. Additionally, there were 20 common DEGs among all three groups (|log_2_FoldChange| > 1 and *p* < 0.05; Figure 5A). These DEGs showed significant differences among the three groups, as demonstrated by the clustering analysis, which revealed aggregated samples from the three groups (Figure 5B). Further analysis using KEGG enrichment revealed that the F3 group was significantly enriched in cellular processes, metabolism, organismal systems, and environmental information processing pathways compared to the F2 group (Figure 5C). Similarly, the F3 group was also significantly enriched in metabolism, cellular processes, organismal systems, environmental information processing, and genetic information processing pathways compared to the F4 group (Figure 5D).

### 3.10. Intestinal Metabolism

The analysis of intestinal metabolites from *H. wyckioides* revealed that certain metabolites, such as L-isoleucine, L-tryptophan, L-proline, PC (P-16:0/2:0), indole-3-acrylic acid, cholic acid, D-PCP, and taurodeoxycholic acid, were primarily enriched in the feces of *H. wyckioides* (Figure 6A). The PLS-DA profile showed distinct clusters for each group, indicating that feeding frequency had a significant impact on the fecal metabolite profile of *H. wyckioides* (Figure 6B). Specifically, the F3 group had significantly higher levels of L-isoleucine, L-valine, and L-proline compared to the F2 group (*p* < 0.05; Figure 6C), whereas the F4 group had significantly higher levels of taurodeoxycholic acid (*p* < 0.05; Figure 6C). In contrast, the F2 and F4 groups had significantly higher levels of D-PCP compared to the F3 group (*p* < 0.05; Figure 6C).

Unannotated and duplicated components were removed to screen for different metabolites (|log_2_FoldChange| > 1 and *p* < 0.05; Figure 7). The F3 group showed 90 different metabolites, with 73 upregulated and 17 downregulated, primarily in the categories of alkaloids, lipids, peptides, and vitamins and cofactors, compared to the F2 group. In contrast, the F4 group only had nine different metabolites, with one upregulated and eight downregulated, mainly in the category of lipids (Figure 7). Furthermore, the F4 group had 65 different metabolites compared to the F3 group, with 11 upregulated and 54 downregulated, primarily in the categories of alkaloids, lipids, hormones and transmitters, peptides, and vitamins and cofactors.

The KEGG metabolic pathways of the differentially abundant metabolites were analyzed by setting a minimum abundance threshold of 0.1% for recognizable tertiary pathways. The results indicated that these metabolites were annotated to 7 KEGG secondary pathways and 17 KEGG tertiary pathways (Figure 8). In comparison to the F2 group, the F3 group showed significant activation of arginine and proline metabolism; arginine biosynthesis; glycine, serine, and threonine metabolism; clavulanic acid biosynthesis; monobactam biosynthesis; nicotinate and nicotinamide metabolism; and pyrimidine metabolism. Meanwhile, the results show significantly inhibited histidine metabolism, glycerophospholipid metabolism, and porphyrin metabolism. In contrast, the F4 group showed significant activation of tryptophan metabolism. The F4 group showed significant inhibition of arginine and proline metabolism, arginine biosynthesis, clavulanic acid biosynthesis, monobactam biosynthesis, arachidonic acid metabolism, nicotinate and nicotinamide metabolism, porphyrin metabolism, and pyrimidine metabolism compared to the F3 group (Figure 8).

## 4. Discussion

### 4.1. Effects of the Feeding Frequency on the Physiological and Biochemical Indices of H. wyckioides

The optimal feeding frequency is crucial for determining growth performance. However, limited studies have explored its effects on juvenile *H. wyckioides*. This study indicated that feeding juvenile *H. wyckioides* three times per day effectively improved their SGR and WGR. Similar results have been found in other aquatic animals, such as Asian seabass [23], juvenile Dolly Varden char (*Salvelinus malma*) [24], and juvenile large yellow croaker (*Pseudosciaena crocea*) [25]. It is possible that feeding frequency may influence growth by altering digestion and absorption [10]. Our results showed that increasing feeding frequency did not significantly affect the moisture, crude protein, crude lipids, or crude ash contents of the fish, which is consistent with findings for Atlantic salmon (*Salmo salar*) [26]. These results suggest that the composition of juvenile *H. wyckioides* is not significantly affected by feeding frequency.

Blood biochemical metrics provide valuable insights into the physiological conditions of fish and are frequently used to examine the impacts of feed nutrition and feeding strategies [27,28]. In this study, we found that feeding frequency has a significant influence on plasma parameters. Specifically, higher feeding frequency was associated with increased levels of serum TC and TG, indicating that excessive feeding can lead to carbohydrate and lipid accumulation in the fish [10]. HDL-C plays a crucial role in the breakdown and metabolism of lipids, reducing lipid accumulation [29]. Furthermore, increases in TP and ALB levels contribute to improving the body’s immune system function [30,31]. Our results showed that feeding *H. wyckioides* three times per day resulted in higher levels of HDL-C, TP, and ALB, suggesting that an optimal feeding frequency can enhance lipid metabolism and immunity. This finding is consistent with previous studies on largemouth bronze gudgeons (*Coreius guichenoti*) [32].

The use of antioxidants has been shown to improve the tolerance of aquatic animals to oxidative stress and liver damage [33]. In this study, the activity of ALT was significantly higher in fish that were administered feed at a frequency of 3 times per day compared to other groups. This suggests that ALT plays a crucial role in amino acid synthesis and metabolism [34]. Additionally, the activities of antioxidant enzymes (GSH-PX, AST, and SOD) were significantly elevated in *H. wyckioides* fed at a frequency of four times per day. This was accompanied by a significant increase in liver MDA levels. However, the expression of the antioxidant genes in the liver was not affected by the feeding frequency. This is probably due to the low expression levels of these genes or the great differences between individuals in each group (Figure 2). These findings indicate that excessive feeding can lead to oxidative stress and lipid deposition in the liver of *H. wyckioides*. Similar research has also been found in other aquatic animals such as juvenile oriental river prawns (*Macrobrachium nipponense*) [35], largemouth bass (*Micropterus salmoides*) [36], and juvenile red swamp crayfish (*Procambarus clarkii*) [33].

Digestive enzymes play a crucial role in promoting the utilization of protein and lipid [37]. In our study, we observed a significant increase in trypsin activity when the feeding frequency was increased to three times per day. Although Su et al. [38] reported that a feeding frequency of three times per day led to an increase in MT and VL, resulting in improved nutrient absorption accelerated growth, our results indicated that the MT and VL were not detected any significant difference among the groups (*p* > 0.05; Table 5). However, excessive feeding frequency can negatively impact the activity of digestive enzymes. Therefore, it is important to maintain an appropriate feeding frequency to promote optimal digestive function and improve production performance through enhanced nutrient absorption.

### 4.2. Effects of the Feeding Frequency on the Intestinal Microbiota of H. wyckioides

The intestinal microbiota plays a significant role in the health of the host, causing disruptions in the host’s immune and metabolic functions [39]. In a study on *H. wyckioides*, it was found that feeding the fish three times per day resulted in an increase in the abundance of Proteobacteria, a major group of bacteria known for their efficient decomposition and utilization of proteins [40]. Additionally, the beneficial bacterium *Clostridium* has been shown to improve the immune function of fish intestines [41]. The higher abundance of *Clostridium* in the F3 group, compared to the other groups, suggests that the intestines of these fish were healthier. On the other hand, the harmful bacterium *Citrobacter*, which poses a threat to fish [42], showed decreases in abundance of 38% and 81% in the F3 group compared to the F2 and F4 groups, respectively. Previous studies have shown that infection with *Citrobacter* can cause damage to the intestinal and liver metabolism of Chinese sturgeon (*Acipenser sinensis*) [43], decreased protein levels and liver degeneration in silver catfish (*Rhamdia quelen*) [44], and decreased total protein levels in the serum of sturgeon [45]. These findings suggest that *Citrobacter* affects both the protein levels and metabolic capacity in the host. Both Proteobacteria and *Citrobacter* can affect the metabolism and utilization of protein, as seen in the serum total protein levels, indicating that feeding frequency may influence protein levels.

*Plesiomonas* is beneficial for the decomposition of carbohydrates and the absorption of nutrients in aquatic animals [46]. The F3 group showed the highest relative abundance of *Plesiomonas*, indicating that this group had the greatest intestinal digestion and absorption abilities.

As a beneficial bacterium, *Clostridium* can improve the immune function of fish intestines [41]. The feeding frequency of three times per day increased the abundance of *Clostridium*. *Lactococcus* is harmful to many aquatic organisms. Infection with *Lactococcus* can lead to a 70–100% incidence of septicemia in the cultivation of Nile tilapia (*Oreochromis niloticus*) [47]. Infection of rainbow trout (*Oncorhynchus mykiss*) with *Lactococcus* can lead to systemic inflammation [48]. Infection of *O. niloticus* can result in outbreaks of *Lactococcus*, leading to death within 48 h after infection [49]. The F3 group presented the lowest relative abundance of *Lactococcus*, decreasing by 89% and 88% compared to the F2 and F4 groups, respectively, indicating that the F3 group had a better health status. These findings suggest that the feeding frequency affected the abundance of *Clostridium* and *Lactococcus* and improved the immune ability at a feeding frequency of three times per day for *H. wyckioides*.

### 4.3. Effects of the Feeding Frequency on the Intestinal Transcriptome of H. wyckioides

Enriched KEGG pathways revealed that the F3 group was significantly enriched in antigen processing and presentation, the chemokine signaling pathway, and the NOD-like receptor signaling pathway compared to the F2 group. The antigen processing and presentation pathway is the cornerstone of adaptive immunity [50]. The chemokine signaling pathway and NOD-like receptor signaling pathway are related to immunity, and in *Acipenser dabryanus*, these genes are also affected by the feeding frequency [9]. Intestinal transcriptome indicated that at a feeding frequency of three times per day enriched immune pathways, which regulate intestinal health.

### 4.4. Effects of the Feeding Frequency on the Intestinal Metabolome of H. wyckioides

Metabolomics can explore alterations in metabolites and pathways within biological systems, providing insights into metabolic processes [51]. Cofactors play a crucial role in regulating oxidative stress [52]. Vitamins are important coenzymes and catalysts in the body [53]. Vitamin C, vitamin E, and carotenoids combat oxidative stress as antioxidants [54]. In our study, the feeding frequency of three times per day significantly increased the levels of metabolites such peptides, vitamins, and cofactors in *H. wyckioides*. Therefore, the feeding frequency might influence oxidative stress in *H. wyckioides*.

Our study also revealed that the pathways significantly activated by feeding three times per day were primarily related to amino acid metabolism, the biosynthesis of other secondary metabolites, and the metabolism of cofactors and vitamins. Amino acid metabolism is closely related to growth, development, and health in fish [55], which is consistent with research on *Acipenser dabryanus* [9]. The metabolism of cofactors and vitamins can alleviate oxidative stress. Our results demonstrated that the feeding frequency can affect the oxidative stress response in *H. wyckioides*, which is consistent with studies of tiger pufferfish (*Takifugu rubripes*) [56], Dolly Varden char (*Salvelinus malma*) [24], and largemouth bronze gudgeon (*Coreius guichenoti*) [32].

## 5. Conclusions

Feeding frequency affects the growth rate of *H. wyckioides* in several aspects, such as the intestinal microbiota, intestinal transcriptome, and metabolism. A feeding frequency of three times per day increased the abundance of *Clostridium* and decreased the abundance of *Lactococcus*, as well as activating multiple immune pathways, such as antigen processing and presentation, the chemokine signaling pathway, and the NOD-like receptor signaling pathway. Additionally, feeding fish three times per day significantly enriched metabolic pathways and the metabolomics that are closely related to their growth. Feeding three times per day increased the abundance of *Plesiomonas*, enhancing nutrient absorption efficiency and accelerating growth. In summary, feeding three times per day improved the growth rate of *H. wyckioides* by enhancing intestinal immunity, promoting intestinal metabolism, and enhancing intestinal digestive function.

## Figures and Tables

**Figure 1 animals-15-01621-f001:**
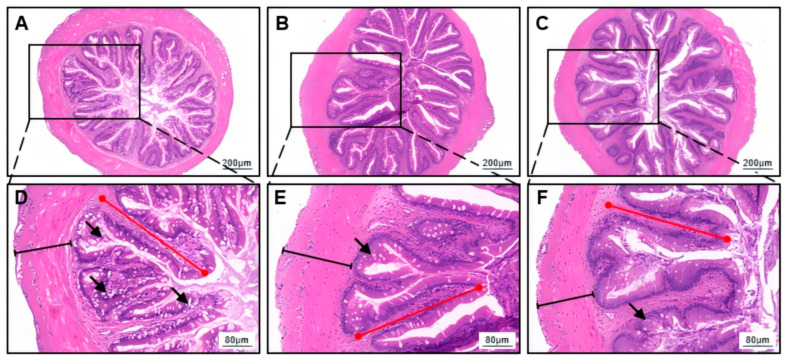
Effects of different feeding frequencies on the intestinal morphology of juvenile *H. wyckioides*. The intestinal sections of *H. wyckioides* from the F2 (**A**,**D**), F3 (**B**,**E**), and F4 (**C**,**F**) groups were examined to determine the effects of different feeding frequencies on the fish intestinal morphology. F2, the fish were fed twice per day at 8:00 and 20:00; F3, the fish were fed three times per day at 8:00, 14:00, and 20:00; F4, the fish were fed four times per day at 9:00, 12:00, 16:00, and 20:00. The black arrows indicate goblet cells, the red lines represent villus height, and the black lines represent muscle thickness.

**Figure 2 animals-15-01621-f002:**
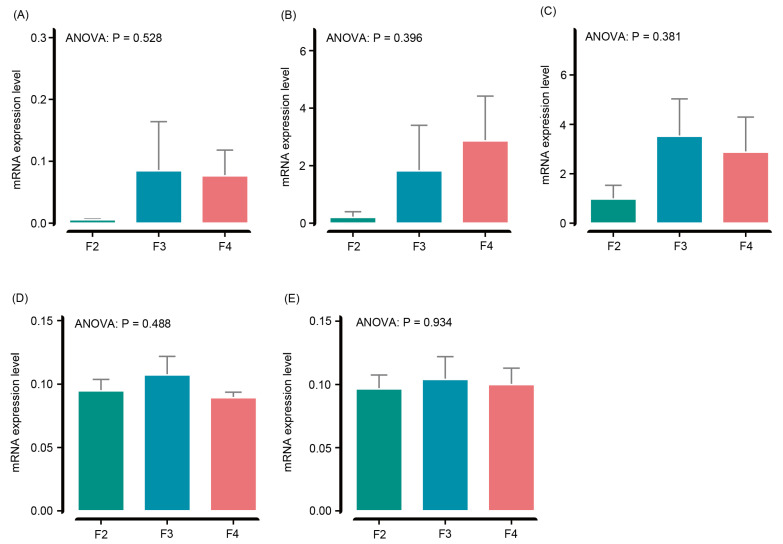
Effects of the feeding frequency on the mRNA expression levels of antioxidant genes in the liver of juvenile *H. wyckioides*. F2, the fish were fed twice per day at 8:00 and 20:00; F3, the fish were fed three times per day at 8:00, 14:00, and 20:00; F4, the fish were fed four times per day at 9:00, 12:00, 16:00, and 20:00. (**A**), Glutathione peroxidase (*gpx*); (**B**) superoxide dismutase (*sod*); (**C**) catalase (*cat*); (**D**) Nuclear factor erythroid 2-related factor 2 (*nrf2*); (**E**) Kelch like ECH associated protein 1 (*keap1*).

**Figure 3 animals-15-01621-f003:**
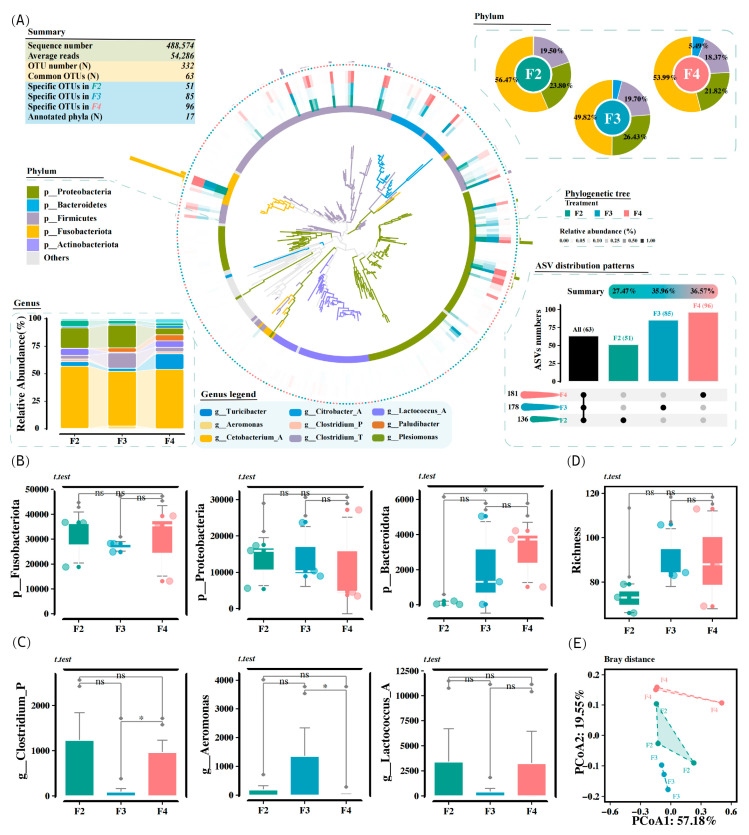
Effects of the feeding frequency on the diversity of the intestinal microbiota of *H. wyckioides*. (**A**) Bar charts explain the relative abundances of dominant taxa at amplicon sequence variants (ASVs) and phylum levels. The upset plot shows the number of unique and shared ASVs in each group. Histogram shows the relative abundances of dominant genera. The phylogenetic tree was drawn using ggtree. The inner to outer frontal layout shows the microbial evolutionary branches colored by different phyla, microbial rings colored by different phyla, and relative abundance heatmaps colored by group. (**B**) Boxplot shows the relative abundance of Bacteroidota. (**C**) Boxplot shows the relative abundance of *Aeromonas*. (**D**) Boxplot of richness. (**E**) PCoA profile based on Bray–Curtis distance. F2, the fish were fed twice per day at 8:00 and 20:00; F3, the fish were fed three times per day at 8:00, 14:00, and 20:00; F4, the fish were fed four times per day at 9:00, 12:00, 16:00, and 20:00. * *p* < 0.05.

**Figure 4 animals-15-01621-f004:**
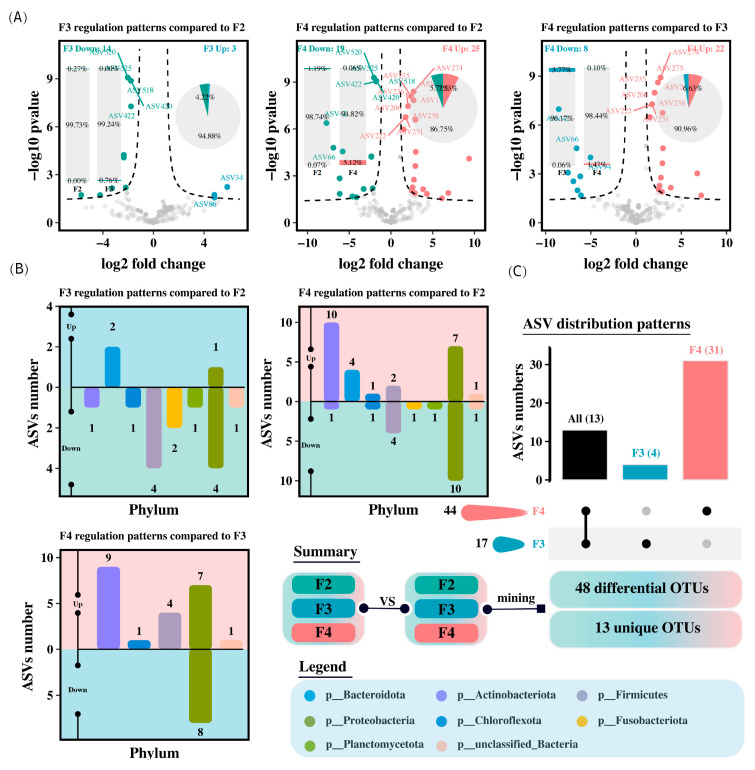
Differential abundances of intestinal microbiota amplicon sequence variants (ASVs) in *H. wyckioides*. (**A**) Differences in ASVs between groups. (**B**) Distribution of differentially abundant ASVs at the phylum level. (**C**) Distribution of differentially abundant ASVs at the species level. F2, the fish were fed twice per day at 8:00 and 20:00; F3, the fish were fed three times per day at 8:00, 14:00, and 20:00; F4, the fish were fed four times per day at 9:00, 12:00, 16:00, and 20:00.

**Figure 5 animals-15-01621-f005:**
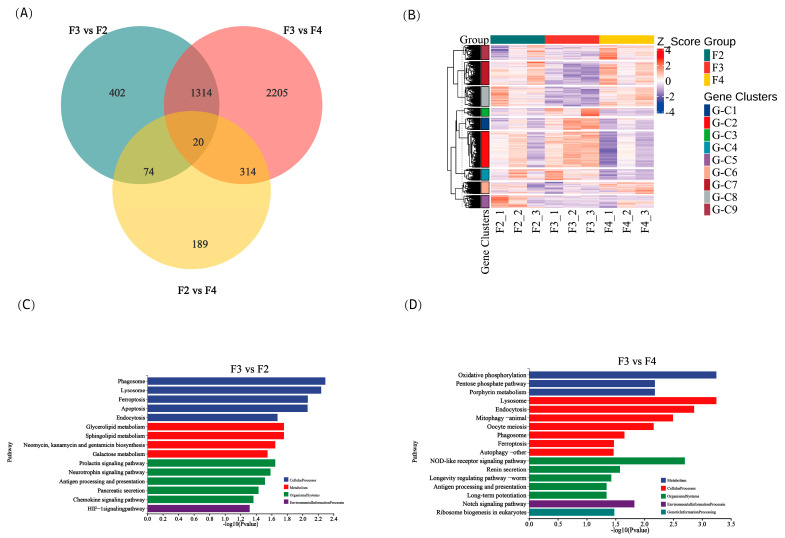
Statistics of differentially expressed genes (DEGs) in the intestine of juvenile *H. wyckioides* fed at different frequencies. (**A**) The numbers of unique and shared DEGs between the groups are presented in a Venn diagram. (**B**) Gene clustering heatmap profile of samples. (**C**) The most enriched KEGG pathways in the F3 group compared to the F2 group. (**D**) The most enriched KEGG pathways in the F3 group compared to the F4 group. F2, the fish were fed twice per day at 8:00 and 20:00; F3, the fish were fed three times per day at 8:00, 14:00, and 20:00; F4, the fish were fed four times per day at 9:00, 12:00, 16:00, and 20:00.

**Figure 6 animals-15-01621-f006:**
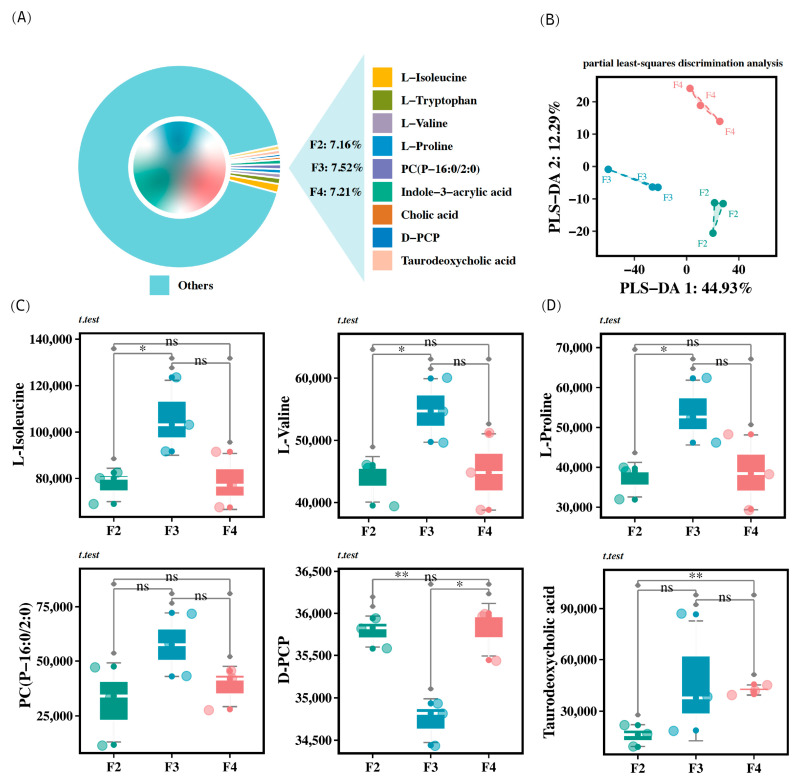
Statistics of differential metabolites in the intestine of juvenile *H. wyckioides* fed at different frequencies. (**A**) The main enriched metabolites in the feces of *H. wyckioides*, along with the proportions of these major metabolites in each feeding frequency group. (**B**) PLS-DA profile of metabolites identified in positive- and negative-ion modes. (**C**,**D**) Significant changes in major metabolites. F2, the fish were fed twice per day at 8:00 and 20:00; F3, the fish were fed three times per day at 8:00, 14:00, and 20:00; F4, the fish were fed four times per day at 9:00, 12:00, 16:00, and 20:00. * *p* < 0.05; ** *p* < 0.01.

**Figure 7 animals-15-01621-f007:**
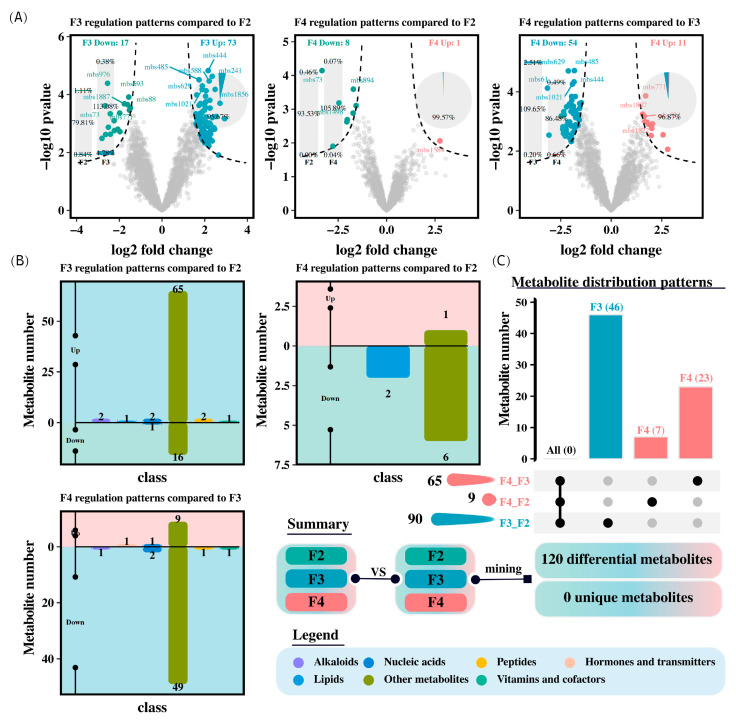
Differentially abundant metabolites in the intestine of *H. wyckioides*. (**A**) Differences in metabolites between groups. (**B**) Pairwise comparison of differentially abundant metabolites between different groups. (**C**) Comparison of differentially abundant metabolites among the three groups. F2, the fish were fed twice per day at 8:00 and 20:00; F3, the fish were fed three times per day at 8:00, 14:00, and 20:00; F4, the fish were fed four times per day at 9:00, 12:00, 16:00, and 20:00.

**Figure 8 animals-15-01621-f008:**
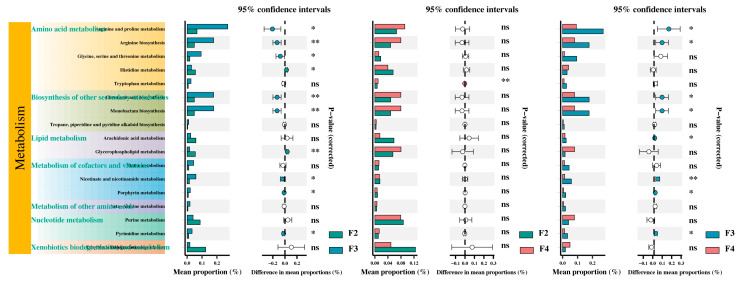
KEGG pathway analysis of the metabolites. F2, the fish were fed twice per day at 8:00 and 20:00; F3, the fish were fed three times per day at 8:00, 14:00, and 20:00; F4, the fish were fed four times per day at 9:00, 12:00, 16:00, and 20:00. * *p* < 0.05; ** *p* < 0.01; ns *p* ≥ 0.05.

**Table 1 animals-15-01621-t001:** Effects of the feeding frequency on growth, physical indices, and whole-body composition of juvenile *H. wyckioides*. F2, the fish were fed twice per day at 8:00 and 20:00; F3, the fish were fed three times per day at 8:00, 14:00, and 20:00; F4, the fish were fed four times per day at 9:00, 12:00, 16:00, and 20:00. Values in the same row with different small superscript letters are significantly different (one-way ANOVA, *p* < 0.05).

Parameter ^1^	F2	F3	F4	*p* Value
FCR	1.73 ± 0.142 ^ab^	1.48 ± 0.111 ^b^	1.85 ± 0.181 ^a^	0.048
SGR (%/d)	1.47 ± 0.002 ^ab^	1.67 ± 0.002 ^a^	1.37 ± 0.003 ^b^	0.044
WGR (%)	114.17 ± 0.191 ^ab^	136.35 ± 0.093 ^a^	102.22 ± 0.071 ^b^	0.043
SR (%)	99.33 ± 0.011	99.33 ± 0.013	99.33 ± 0.014	0.999
HSI (%)	1.81 ± 0.002	1.96 ± 0.001	1.91 ± 0.003	0.709
VSI (%)	11.82 ± 0.021	11.46 ± 0.013	11.69 ± 0.021	0.878
CF (g/cm^3^)	0.01 ± 0.001	0.01 ± 0.002	0.01 ± 0.001	0.531
Moisture (%)	69.78 ± 0.001	69.90 ± 0.012	69.89 ± 0.002	0.942
Crude protein (%)	17.82 ± 0.013	17.67 ± 0.001	17.81 ± 0.012	0.901
Crude lipid (%)	8.08 ± 0.003	8.53 ± 0.009	8.15 ± 0.002	0.548
Crude ash (%)	2.64 ± 0.003	2.79 ± 0.002	2.72 ± 0.003	0.713

^1^ WGR, weight gain rate; SGR, specific growth rate; FCR, feed conversion ratio; SR, survival rate; HSI, hepatosomatic index; VSI, viscerosomatic index; CF, condition factor.

**Table 2 animals-15-01621-t002:** Effects of the feeding frequency on the serum biochemical indices of juvenile *H. wyckioides*. Data are presented as the means and standard errors of the means. F2, the fish were fed twice per day at 8:00 and 20:00; F3, the fish were fed three times per day at 8:00, 14:00, and 20:00; F4, the fish were fed four times per day at 9:00, 12:00, 16:00, and 20:00. Values in the same row with different small superscript letters are significantly different (one-way ANOVA, *p* < 0.05).

Parameter ^1^	F2	F3	F4	*p* Value
TC (mmol/L)	2.84 ± 1.034 ^b^	3.30 ± 0.791 ^b^	4.54 ± 0.971 ^a^	0.023
TG (mmol/L)	12.20 ± 1.634	14.05 ± 2.821	14.42 ± 0.472	0.371
HDL-C (mmol/L)	3.09 ± 0.574 ^b^	4.37 ± 0.252 ^a^	3.41 ± 0.662 ^ab^	0.049
LDL-C (mmol/L)	5.26 ± 0.431	5.06 ± 0.594	5.61 ± 0.374	0.406
GLU (mmol/L)	4.64 ± 0.205	5.04 ± 0.835	4.20 ± 0.342	0.253
TP (g/L)	29.82 ± 4.306	32.88 ± 1.338	28.93 ± 2.471	0.301
ALB (g/L)	13.42 ± 1.711	14.82 ± 3.213	13.05 ± 2.312	0.673

^1^ TC, total cholesterol; TG, triglyceride; HDL-C, high-density lipoprotein cholesterol; LDL-C, low-density lipoprotein cholesterol; GLU, glucose; TP, total protein; ALB, albumin.

**Table 3 animals-15-01621-t003:** Effects of the feeding frequency on the antioxidant capacity in the liver of juvenile *H. wyckioides*. Data are presented as the means and standard errors of the means. F2, the fish were fed twice per day at 8:00 and 20:00; F3, the fish were fed three times per day at 8:00, 14:00, and 20:00; F4, the fish were fed four times per day at 9:00, 12:00, 16:00, and 20:00. Values for each item with different superscripts are significantly different (one-way ANOVA, *p* < 0.05).

Parameter ^1^	F2	F3	F4	*p* Value
T-AOC (mmol/g prot)	0.15 ± 0.023 ^b^	0.17 ± 0.021 ^b^	0.23 ± 0.042 ^a^	0.026
MDA (nmol/mg prot)	0.75 ± 0.092 ^b^	0.60 ± 0.101 ^b^	0.97 ± 0.151 ^a^	0.001
SOD (U/mg prot)	7.78 ± 1.661 ^b^	6.60 ± 0.713 ^b^	10.15 ± 0.883 ^a^	0.025
GSH-PX (U/mg prot)	34.74 ± 4.892 ^b^	35.29 ± 7.421 ^b^	55.62 ± 12.412 ^a^	0.044
ALT (U/g prot)	3.46 ± 0.511 ^b^	9.36 ± 0.472 ^a^	3.77 ± 1.031 ^b^	0.001
AST (U/g prot)	42.31 ± 5.361 ^b^	71.17 ± 24.592 ^ab^	77.83 ± 3.492 ^a^	0.053
ALP (King unit/g prot)	1.62 ± 0.734 ^a^	0.46 ± 0.291 ^b^	2.22 ± 0.221 ^a^	0.004

^1^ T-AOC, total antioxidant capacity; MDA, malondialdehyde; SOD, superoxide dismutase; GSH-PX, glutathione peroxidase; ALT, alanine transaminase; AST, aspartate aminotransferase; ALP, alkaline phosphatase.

**Table 4 animals-15-01621-t004:** Effects of the feeding frequency on the intestinal digestive enzyme activities of *H. wyckioides*. Data are presented as the means and standard errors of the means. F2, the fish were fed twice per day at 8:00 and 20:00; F3, the fish were fed three times per day at 8:00, 14:00, and 20:00; F4, the fish were fed four times per day at 9:00, 12:00, 16:00, and 20:00. Values for each item with different superscripts are significantly different (one-way ANOVA, *p* < 0.05).

Parameter	F2	F3	F4	*p* Value
trypsin (U/mg prot)	2.24 ± 0.991 ^a^	3.30 ± 0.382 ^b^	4.27 ± 0.611 ^c^	0.004
amylase (U/mg prot)	1.09 ± 0.972	1.19 ± 0.774	1.00 ± 0.621	0.937
lipase (U/g prot)	38.51 ± 8.592 ^b^	84.31 ± 14.819 ^a^	49.50 ± 1.032 ^b^	0.002

**Table 5 animals-15-01621-t005:** Effects of the feeding frequency on the intestinal morphology of juvenile *H. wyckioides*. Data are presented as the means and standard errors of the means. F2, the fish were fed twice per day at 8:00 and 20:00; F3, the fish were fed three times per day at 8:00, 14:00, and 20:00; F4, the fish were fed four times per day at 9:00, 12:00, 16:00, and 20:00.

Parameter ^1^	F2	F3	F4	*p* Value
GCs (number)	31.80 ± 12.261	25.67 ± 9.363	24.57 ± 8.048	0.359
MT (mm)	0.13 ± 0.028	0.16 ± 0.022	0.14 ± 0.032	0.275
VL (mm)	0.39 ± 0.071	0.44 ± 0.072	0.35 ± 0.082	0.122

^1^ GCs, goblet cells; MT, muscularis thickness; VL, villous length.

## Data Availability

The data will be made available upon request. The 16S rRNA sequencing and transcriptome sequencing data from this study have been uploaded to NCBI with accession numbers PRJNA1173319 and PRJNA1172893, respectively. The metabolomics data have been uploaded to Metabolights, with the URL www.ebi.ac.uk/metabolights/MTBLS11417 (accessed on 28 December 2024).

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
