# Peer review of "Feeding Frequency Affects the Growth Performance and Intestinal Health of Juvenile Red-Tail Catfish (Hemibagrus wyckioides) with the Same Amount of Daily Feed"

_animals, 2025, doi:10.3390/ani15111621_

Round 1
Reviewer 1 Report
Comments and Suggestions for Authors
General comments:
The authors compare the effect of feeding frequency on Hemibagrus wyckioides juvenies. They investigate the fishes’ health at great depth, enriching information on growth performance, histomorphology and digestive enzyme abundance with liver gene expression, intestinal transcriptomic and gut content metabolomics. The authors make efficient use of clustering techniques for data exploration. In addition I particularly like that the authors don’t forget the basics over all the omics data: the intestinal histomorphology.
Specific comments:
L78-85: 450 fish were divided in three tanks of 150 fish each in 2x2x1m tanks. Three replicates á 50 fish in each group. How did you divide the tanks so they would house three replicates? Did you have small net cages inside the 2x2x1m tanks? I do not know what it is that you describe as three-dimensional circulating water culture tanks in L79. Could you please clarify?
L 188: did you mean HSI, not HIS? a word autocorrect fail…
L 202: HSI hepatosomatic vs. hepatopancreas index? I think you should decide for one.
Table 3: please keep H. wyckioides italicized throughout the entire manuscript.
Figure 1: Did you observe consistent differences in muscle density?
L465+543 vs. L246+247: thicker muscle layer and villus length in T3? These strong statements in discussion and conclusion are not supported by the results.
Author Response
General comments:
The authors compare the effect of feeding frequency on Hemibagrus wyckioides juvenies. They investigate the fishes’ health at great depth, enriching information on growth performance, histomorphology and digestive enzyme abundance with liver gene expression, intestinal transcriptomic and gut content metabolomics. The authors make efficient use of clustering techniques for data exploration. In addition I particularly like that the authors don’t forget the basics over all the omics data: the intestinal histomorphology.
Response
We sincerely thank you for reviewing our manuscript and providing the valuable comments. These comments are very important for us to correct the mistakes in the text and improve the readability of our manuscript. We have carefully revised our manuscript according to these comments. Thank you again.
Specific comments:
L78-85: 450 fish were divided in three tanks of 150 fish each in 2x2x1m tanks. Three replicates á 50 fish in each group. How did you divide the tanks so they would house three replicates? Did you have small net cages inside the 2x2x1m tanks? I do not know what it is that you describe as three-dimensional circulating water culture tanks in L79. Could you please clarify?
Response
We are sorry that we didn't explain our experimental design clearly. A total of 450 fish were randomly distributed into three groups. Each group contained three tanks with a diameter of 2 meters and a height of 1 meter. Each tank contained 50 fishes. In order to show our experimental device better, we added Figure S1.
Specific comments:
L 188: did you mean HSI, not HIS? a word autocorrect fail…
Response
Yes, we did. HIS should be HSI. We have corrected the spelling mistake there. Thank you for your comment.
Specific comments:
L 202: HSI hepatosomatic vs. hepatopancreas index? I think you should decide for one.
Response
Thank you for your comment. HSI should be hepatosomatic index. We have revised the error.
Specific comments:
Table 3: please keep H. wyckioides italicized throughout the entire manuscript.
Response
We have italicized the H. wyckioides throughout the entire manuscript according to your comment.
Specific comments:
Figure 1: Did you observe consistent differences in muscle density?
Response
Thank you for your comment. It's a pity that we didn't measure muscle density, although different feeding frequencies did not cause significant differences in intestinal tissue structure (Fig. 1 and Table 5).
Specific comments:
L465+543 vs. L246+247: thicker muscle layer and villus length in T3? These strong statements in discussion and conclusion are not supported by the results.
Response
Thank you very much for your comment. We have revised the inappropriate expressions.
Reviewer 2 Report
Comments and Suggestions for Authors
Manuscript Title: Feeding frequency affects the growth performance and intesti-2 nal health of juvenile red-tail catfish (Hemibagrus wyckioides) 3 with the same amount of daily feed
This study aimed to determine the impact of feeding frequency of red-tail catfish (Hemibagrus wyckioides) on serum biochemical parameters, intestinal histology and transcriptome, antioxidant and digestive enzymes capacity, antioxidant related gene expressions and microbiota. In the study, the effect of feeding frequency provided a multi-faceted examination of the mechanism with many analyses.
The results of some analyses are not stated in the abstract. I believe this is due to the word limit of the abstract section in the journal. In this case, I suggest replacing some of the keywords with ones that relate to analyses not mentioned in the abstract.
Materials and Methods
Line 98-99. Is there a reason why blood samples are kept at 4 degrees for 12 hours and then centrifuged after being taken? If so, it should be stated in the text.
Line 128. It should be stated how many sections were taken for each fish and how many villi were measured in those sections.
Results
Some results do not include linear and quadratic regression values ​​in their tables. Were they not added because these analyses were not performed or because no difference was seen? If no difference was seen, it should be stated in the results section with at least one sentence.
Line 271. Why is t-test written on gene expression graphs?
Discussion
Line 457. The results of the study show that there is a difference between the groups in antioxidant enzyme activity in the liver, but there is no difference in gene expression. It would be good if this part is discussed a little more.
Author Response
Comment
Manuscript Title: Feeding frequency affects the growth performance and intesti-2 nal health of juvenile red-tail catfish (Hemibagrus wyckioides) 3 with the same amount of daily feed
This study aimed to determine the impact of feeding frequency of red-tail catfish (Hemibagrus wyckioides) on serum biochemical parameters, intestinal histology and transcriptome, antioxidant and digestive enzymes capacity, antioxidant related gene expressions and microbiota. In the study, the effect of feeding frequency provided a multi-faceted examination of the mechanism with many analyses.
The results of some analyses are not stated in the abstract. I believe this is due to the word limit of the abstract section in the journal. In this case, I suggest replacing some of the keywords with ones that relate to analyses not mentioned in the abstract.
Response
We sincerely thank you for reviewing our manuscript and providing the valuable comments. These comments are very important for us to correct the mistakes in the text and improve the readability of our manuscript. We have carefully revised our manuscript according to these comments. Furthermore, we added some keywords according to your comment. Thank you again.
Comment
Materials and Methods
Line 98-99. Is there a reason why blood samples are kept at 4 degrees for 12 hours and then centrifuged after being taken? If so, it should be stated in the text.
Response
A total of 45 fish were analyzed. Because blood collection takes time, to minimize the influence of storage time before serum biochemical parameters determination on the results and ensure sufficient sedimentation of plasma, blood samples were collected and stored for 12 h at 4 °C. They were then centrifuged at 3000 rpm for 10 minutes at 4 °C to obtain serum. We have stated in the text according to your comment.
Comment
Line 128. It should be stated how many sections were taken for each fish and how many villi were measured in those sections.
Response
Fifteen fish were analyzed in each group (five fish in each tank), and one section was taken for each fish and three villi were measured in each section. We have added the description in our revised manuscript.
Comment
Results
Some results do not include linear and quadratic regression values in their tables. Were they not added because these analyses were not performed or because no difference was seen? If no difference was seen, it should be stated in the results section with at least one sentence.
Response
Thank you very much for your comment. We have deleted the linear and quadratic regression results because we think the results of regression analysis with only three independent variables (feeding frequency) are unreliable.
Comment
Line 271. Why is t-test written on gene expression graphs?
Response
We are very sorry for the mistake caused by typesetting, and we have revised the Figure 2. We are very grateful to you for pointing out the mistakes there.
Comment
Discussion
Line 457. The results of the study show that there is a difference between the groups in antioxidant enzyme activity in the liver, but there is no difference in gene expression. It would be good if this part is discussed a little more.
Response
This is probably due to the low expression levels of these genes or the great differences between individuals in each group, as shown in the Figure 2. We have added the discussion in our revised manuscript according to your comment.